# Cost and Cost-Effectiveness of the Mediterranean Diet: An Update of a Systematic Review

**DOI:** 10.3390/nu16121899

**Published:** 2024-06-16

**Authors:** Corrado Colaprico, Davide Crispini, Ilaria Rocchi, Shizuka Kibi, Maria De Giusti, Giuseppe La Torre

**Affiliations:** Department of Public Health and Infectious Diseases, Sapienza University of Rome, Piazzale Aldo Moro 5, 00185 Rome, Italy; corrado.colaprico@uniroma1.it (C.C.); davidecrispini@gmail.com (D.C.); ilariarocchi16@gmail.com (I.R.); shizuka.kibi@uniroma1.it (S.K.); maria.degiusti@uniroma1.it (M.D.G.)

**Keywords:** Mediterranean diet, cost, cost effectiveness, cost utility, cost benefit, economic evaluation, review, global

## Abstract

It is well known that the Mediterranean diet (DM) is beneficial for health, as years of research globally have confirmed. The aim of this study was to update a previous systematic review that assessed the cost-effectiveness of adherence to the DM as a strategy for the prevention of degenerative diseases by evaluating the economic performance of this diet. The research approach utilized three electronic databases: PubMed, Scopus, and Web of Science. A comprehensive search was conducted to retrieve articles based on a PRISMA-compliant protocol registered in PROSPERO: CRD 42023493562. Data extraction and analysis were performed on all included studies. One thousand two hundred and eighty-two articles were retrieved, and once duplicates and irrelevant articles were removed, fifteen useful articles were reviewed. The studies indicated a clear link between dietary habits, health, and economic aspects related to dietary cost and health spending. Recognizing the significant health benefits associated with adopting DM and the potential savings on health care spending, it is important for national public health programs to consider policies that support this lifestyle.

## 1. Introduction

In recent decades, there has been a progressive increase in the incidence of chronic diseases such as cardiovascular disease (CVD), diabetes, metabolic syndrome, neoplasms, dyslipidemia, high blood pressure, and Alzheimer’s disease [1]. On one hand, this steady increase is secondary to major changes in lifestyle, and on the other hand, it is related to the increase in life expectancy achieved through scientific progress [2,3]. However, research shows that an increase in life expectancy does not correspond with improved quality of life [2,3]. The Mediterranean diet (DM) is universally recognized as a dietary pattern capable of having beneficial long-term health effects [4].

Many researchers have emphasized the beneficial role of the DM for CVD, diabetes, cancer, Alzheimer’s disease, dyslipidemia, and metabolic syndrome [5,6,7]. In addition, it has been widely established that the DM plays an important role in reducing inflammation and the well-being of the gut microbiome [8].

The DM has evolved from original definition; however, many of the same key findings—such as high intakes of extra virgin olive oil, green leafy vegetables, fruits, grains, nuts, and legumes; moderate intakes of fish, dairy products, and wine; and limited intakes of meat, processed meats, and sweets—have been supported [4]. Today, there are also validated indicators that can tell us how “Mediterranean” a meal is, such as the Mediterranean Adequacy Index (MAI) [9].

The United Nations Educational, Scientific, and Cultural Organization (UNESCO) itself has recognized the DM as an intangible cultural heritage of humanity and as a cultural and health model. It is worth mentioning that the DM encompasses not only the dietary aspect, but the broader lifestyle aspect of which physical activity, which is also fundamental in the prevention of chronic diseases, among others, is a part [10].

The Seven Countries Study showed that populations in the Mediterranean region experienced lower CVD mortality than populations in northern Europe or the United States, probably due to different dietary patterns [11]. Today, we have much evidence available that has confirmed the benefits associated with adherence to a Mediterranean dietary pattern on CVD risk factors [7]. Lifestyle changes related primarily to the DM have the potential to change disease outcomes and, therefore, the costs associated with their management [12].

Health care costs attributable to nutrition-related diseases and conditions are substantial. In 2019, the Organization for Economic Co-operation and Development (OECD) countries’ report stated that 8.4 percent of health spending is related to overweight-related conditions, reminding us that every dollar saved to prevent obesity yields six times the economic return [13].

The 2019 World Health Organization (WHO) report reminds us that CVD is the leading cause of death in industrialized countries, with 18.6 million deaths. Diabetes afflicts more than 530 million sufferers worldwide as of 2022 [1]. The WHO reports more than 1 billion obese people in the world according to 2022 data, and the Atlas 2023 report estimates that by 2035, more than half of the world’s population will be obese or overweight, with an economic cost of 4.32 trillion, or about three percent of global GDP [14].

Food choices are influenced by social, cultural, and economic constraints. However, in contrast to the past, recent evidence supports that greater adherence to a Mediterranean dietary pattern is inversely proportional to the cost of groceries [15].

The objective of this study was to conduct an economic evaluation through a systematic review of the scientific literature to assess the cost-effectiveness of adherence to the DM as a strategy for the prevention of degenerative diseases by evaluating the economic performance of this diet, investigating the monetary costs of adopting this dietary pattern and determining the cost differences between low and high adherence to it.

## 2. Materials and Methods

This study stems from the intent to perform an update of a previous systematic review entitled ‘Cost and Cost-Effectiveness of the Mediterranean Diet: Results of a Systematic Review’ [16], published 10 years ago, with the aim of examining and evaluating new evidence concerning the health–economic implications of adhering to the Mediterranean model. This study was carried out according to the Preferred Reporting Items for Systematic Reviews (PRISMA) guidelines [17]. This review was recorded in PROSPERO, the International Prospective Register of Systematic Reviews, and the registration number is CRD 42023493562.

### 2.1. Search Strategy

This review was conducted using three electronic medical journal databases, Scopus, PubMed, and Web Of Science, to search for published studies on the economic evaluation of adherence to the DM. The keywords used for the research were ‘Mediterranean diet’, ‘cost effectiveness’, ‘cost utility’, ‘cost benefit’, and ‘cost’. Combined searches were conducted for: ‘Mediterranean diet AND cost effectiveness’, ‘Mediterranean diet AND cost utility’, ‘Mediterranean diet AND cost benefit’, and “Mediterranean diet AND cost”. A restriction was applied to the date of the database search, starting from 2013 in order to exclude studies analyzed in the previous review. The study selection had no language limitation. The article search and data extraction were carried out between 17 December 2023 and 17 February 2024.

### 2.2. Study Selection

The review process was conducted using a multi-stage approach. Selection and exclusion of duplicates were conducted independently by the authors and managed with ZOTERO, release 6.0.13, a free and open-source reference management software to manage bibliographic data. In the selection process, abstracts were initially read independently by two researchers who identified potentially suitable full-texts, which were then retrieved and evaluated to decide on their final inclusion. Relevant studies in the bibliography of the analyzed articles that contained findings related to the economic evaluation of the DM were also examined and included in this review. If an included publication was unavailable as full text in English, the corresponding author was contacted to verify whether the eligibility criteria were met. Discrepancies and disagreements were discussed and resolved through a consensus session with a third-party researcher.

### 2.3. Inclusion and Exclusion Criteria

Articles were reviewed and included if (a) the study was based on the assessment of the impact of adherence to the DM on the epidemiology of CVD or other chronic diseases; (b) the study assessed the impact of adopting a DM on individual dietary costs and monetary costs for Mediterranean food intake patterns (e.g., cross-sectional survey based on the assessment of monetary costs for all foods in the Food Frequency Questionnaire (FFQ) administered, calculated by multiplying the amount of food consumed by the FFQ by national average prices, etc.). Articles were excluded if (a) studies did not refer to the DM in the results of the economic evaluation; (b) economic data were not reported.

### 2.4. Data Extraction

A data collection form was developed to confirm the relevance of the studies and to extract their characteristics. Data extraction was conducted independently by reviewers, extracting data from all included studies. The following information was extracted from the studies: name of first author, title of article, country of first author, year of publication, study design, type of study, main outcome, study objective, country in which the study was conducted, sample, efficacy/cost measures, main results, and quality assessment. To ensure accurate data collection, the extracted data were compared independently by each reviewer. Discrepancies and disagreements were discussed and resolved through a consensus session with a third-party researcher.

### 2.5. Quality Assessment

A quality assessment of the observational studies was conducted using the Newcastle–Ottawa Scale (NOS) [18]. This is a validated, easy-to-use scale of 8 items in three domains: selection, comparability, and exposure/outcome for case–control or cohort studies, respectively. Each item can be assigned one point, with the exception of comparability, which has the potential to score up to two points. Studies are rated from 0 to 9, with studies rating either 0–3 (poor quality), 4–6 (fair quality), or 7–9 (good/high quality). The NOS scale adapted for cross-sectional studies was used to assess the quality of cross-sectional studies [19]. The Jadad scale was used for the evaluation of randomized clinical trials (RCTs) [20] and, finally, Drummond’s original checklist modified by La Torre et al. was used for economic evaluation studies [21]. Drummond’s checklist is composed of 35 items divided into 3 sections: study design, data collection, and analysis and interpretation of results. With the aim of assigning a score to each item, a group of experts was asked to attribute a score according to its importance. The weighted scores that were assigned by consensus to the study design, data collection, and analysis and interpretation of results were 26, 45, and 48, respectively. For each item section, the maximum achievable score was as follows: 1. study design (7 items): maximum global score = 26; 2. data collection (14 items): maximum global score = 45; 3. analysis and interpretation of results (14 items): maximum global score = 48. When the item was not applicable to the study, we reduced the maximum global score from the relative weighted score item.

## 3. Results

### 3.1. Identification of Relevant Research

By including only studies published from 2013, but without language restrictions, we found a total of 347 articles from the PubMed search, 455 articles from the Scopus search, and 480 articles from the Web of Science search.

After removing duplicates, a total of 818 articles were found. We included all study designs, including systematic reviews, from which we extracted all relevant bibliographic references.

In addition, the articles were carefully screened for both title and abstract obtained from the three search engines separately and duplicates removed in each database.

Out of 818 articles, 464 were duplicates, 772 were removed because they were irrelevant (lack of data, they were a search for more data on the results of the DM, or simply because they did not correspond to our objective). Finally, 46 articles were assessed for eligibility: all full texts were examined; at the end of the assessment, 32 were excluded because they lacked data on economic evaluation or did not have adequate full texts. Among the references of the 14 systematic reviews found, only one was consistent with the objective of our research, and from this, an RCT study by Lara et al. was included. At the end of the selection process, only 15 articles met the predetermined criteria described above. Search criteria are summarized in Figure 1.

The included studies had two different objectives: (a) one type of study was based on the analysis of cost-effectiveness or the economic impact of adherence to the DM on the epidemiology of cardiovascular or other chronic diseases; (b) the other type of study included information about the direct and indirect costs derived from the rate of adherence to a Mediterranean lifestyle (Table 1 and Table 2).

### 3.2. Quality Assessment

The quality assessment was conducted using the Newcastle-Ottawa scale (NOS) [18] for observational studies and the Jadad scale [20] for RCT studies, whose results are shown in Table 1. For the economic evaluation of the studies, the original Drummond checklist modified by La Torre et al. [21] was used; the results are summarized in Table 2. The highest score was obtained by Segal et al. with 104/116 (89.65%), the lowest score 95/116 (81.89%) was awarded to Schepers and Annemans, while an intermediate score was assigned to Jones et al. 97/116 (83.62%). The average quality of the three articles was high (85.38%).

### 3.3. Description of Results

The study performed by Lara et al. (2015) [22] presented two research studies conducted among elderly residents in northeast England. The first study investigated the validity and clarity of two different ways to graphically represent the DM, namely, a pyramid and a plate. The second study evaluated the feasibility and acceptability of a brief intervention aimed at promoting the DM, differentiated into two levels of dietary advice, and examined dietary cost in relation to DM adherence during the intervention. In the first study, participants completed a questionnaire regarding demographics and health information, and were asked to keep a three-day food diary before and after an interactive group educational session on the DM. The analysis revealed that there were no significant differences in the preference for a plate or pyramid mode of graphical representation of the DM among elderly Britons. In addition, the majority of participants showed moderate adherence to DM. The second study was 3 weeks in duration with two intervention arms. Group 1 participated in a group educational session on DM and were encouraged to adopt it for three weeks, while group 2 received additional support from the research team during the intervention period. Both groups showed improvements in eating patterns, with increased intake of healthy foods and a reduction in meat consumption, without incurring significantly higher costs. In summary, both studies demonstrated that DM represents an acceptable pattern of healthy eating for the British elderly, and that a brief dietary intervention aimed at promoting DM is highly acceptable and can lead to improvements in eating patterns without incurring significantly higher costs.

The study by Jones et al. (2019) [23] highlighted the pressing health and economic burden from diet-related non-communicable disorders, such as obesity, type 2 diabetes, CVD, and cancer, on a global scale. CVD, in particular, is the most prevalent and costly cause of mortality worldwide, with an estimated cost of USD 863 billion in 2010, projected to exceed USD 1 trillion by 2030. Eating habits consistent with the principles of a healthy diet can promote individual and societal well-being, generating significant economic benefits. The study employed a Monte Carlo simulation to assess the impacts of adherence to the DM on reducing health care costs associated with CVD in the United States and Canada in the short and long term. The results showed significant short-term and long-term accumulated annual savings from increasing DM adherence, emphasizing the importance of public policies in promoting healthy food choices. Specifically, in the United States, annual savings were estimated to range from USD 8.2 billion to USD 31 billion, and in Canada from USD 320 million to USD 1.2 billion, depending on the degree of DM adherence. In the long term, cumulative savings over a 10-year period were projected to range from USD 157.1 billion to USD 596.1 billion in the United States and USD 5.5 billion to CAD 20.9 billion in Canada.

Lampropoulos et al. (2020) [24] examined the impact of the DM on hospital length of stay, long-term mortality, and inpatient-related financial cost for hospitalized elderly patients. The results revealed that each unit increase in DM adherence score was associated with an average reduction in hospital length of stay of 0.3 days. In addition, for each unit increase in albumin, an indirect indicator of adequate nutritional status, the length of hospital stay was reduced by 2.1 days. Financially, it was found that the additional cost of hospital stay was related to the extended length of stay, with an average increase of about 112 euros for each additional day of hospitalization. A higher level of DM adherence was associated with a reduction in cost, with an average decrease of 40 euros for each unit increase in DM adherence score. Finally, a reduction in long-term mortality risk was observed to be significantly related to DM adherence, with a 13 percent decrease in the risk of death for each unit increase in DM adherence score. These results underscore the importance of the DM in hospital management of elderly patients, highlighting the benefits in both health and financial costs.

The study conducted by Segal et al. (2020) [25] evaluated the effectiveness and efficiency of a dietary intervention for the treatment of major depression. Depression has a negative impact on quality of life and personal, social, and economic costs. Currently, the most common treatments for depression are pharmacotherapy and psychotherapy. However, some studies suggest that a healthy diet could reduce the risk of depression and that a dietary intervention could be effective in treating depression. The study, called HELFIMED, recruited adults with major depression and randomly assigned them to either a group dietary program or a social program. The dietary group received counseling on the DM, followed by group cooking meetings and food supplementation, while the social group participated in social meetings. Both groups showed significant improvements in depressive symptoms, but the dietary group achieved greater improvements. Improvements were maintained even at 6-month follow-up. The economic analysis evaluated the cost to gain one healthy life year (QALY) and the cost per case of resolved depression. The dietary intervention was found to be highly cost-effective, with a cost per QALY of about EUR 1775, well below the threshold considered acceptable in health care. In addition, the cost per case of depression treated was found to be about EUR 1120. In conclusion, group dietary intervention has proven to be an outstanding social investment for the treatment of major depression. It could be an effective and cost-effective strategy for dealing with depression and improving the quality of life of people with this condition.

Seconda et al. (2017) [26] analyzed the impact of DM and organic food consumption on sustainability, integrating sociocultural, economic, nutritional, and environmental aspects. Using data from a large sample from the NutriNet-Santé study, the study compared four distinct groups of subjects based on their level of organic food consumption and adherence to the DM. The results of the analysis indicate that participants who followed the DM or adopt a diet based on organic food consumption showed better indicators of sustainability in terms of health, nutrition, environmental impact, and purchase motivation. However, it appears that the cost of the diet was significantly higher for organic food consumers compared to non-consumers. Specifically, the average daily diet cost for participants following the DM was EUR 9.11, while for those consuming organic food was EUR 10.90. The share of the food budget appears to be higher among organic food consumers, with 26.4 percent of the budget allocated to food in the group combining DM and organic food consumption (Org-Med). In addition, those who combine both dietary patterns show the best results in terms of economic and environmental sustainability. These results suggest that considering sustainable dietary patterns is crucial in the transition to a more sustainable diet, with significant implications for public health and conservation of environmental resources.

Tong et al. (2018) [27] focused on the association between adherence to the DM and food cost in a large British population, carefully exploring the role of socioeconomic factors in this correlation. Using data from the Fenland study, a cohort including 12,435 individuals in the United Kingdom, the researchers employed a specific score to assess adherence to the DM through an analysis of food questionnaires and food costs based on market prices of products. The results highlight an association between a high degree of adherence to the DM and a slight but significant increase in dietary costs, which appears to be differentially influenced by the socioeconomic position of participants. Specifically, subjects with lower socioeconomic status show a stronger correlation between adherence to the DM and dietary costs than participants with higher socioeconomic status. The analysis showed that an average increase in daily costs of about EUR 0.23 (95% CI 0.19, 0.28) correlated with high adherence to the DM.

Yacoub Bach et al. (2023) [28] analyzed six different diets in order to assess their environmental impact, nutritional quality, and associated economics. Through an in-depth integrative analysis of data from various academic sources, substantial differences between the diets were examined. From an environmental perspective, it was found that diets with a higher proportion of foods of animal origin show a greater environmental impact, characterized by high greenhouse gas emissions, a larger land use extent, greater eutrophication potential, and higher water withdrawal. In contrast, plant-based diets mainly tended to have a lower environmental footprint on these indicators. In terms of nutritional quality, diets with a greater variety of foods and a lower presence of animal products showed a superior nutritional composition, with a higher intake of nutrients considered beneficial to health. However, there is significant variation in nutritional quality among the different diets, with some characterized by a higher SAIN/LIM ratio than others, indicating greater nutritional fitness. SAIN/LIM classifies foods into four classes based on two scores: a nutrient density score (NDS) called SAIN and a score of nutrients to limit called LIM, and one threshold on each score [36]. Economic analysis revealed considerable diversity in the costs associated with the different diets, with some diets found to be cheaper than others. Diets high in plant foods tended to be more affordable, while those with a higher presence of animal products appeared to be more expensive. In summary, the analysis conducted highlights the complex interactions between diet, environmental sustainability, and human well-being, providing a comprehensive overview of the implications of different food choices on the planet, health, and the economy.

Razavi et al. (2020) [29] examined the impact of a culinary educational intervention focused on the DM within families with children in the Greater New Orleans region through a randomized controlled trial. The experimental group participated in a 12 h training program focused on nutrition education and culinary preparation, while the control group received standard nutrition education. The results revealed a significant improvement in adherence to the DM in the experimental group, as measured by a validated DM adherence score, compared with the control group. This improvement was associated with estimated economic savings of at least EUR 20 per week per household in the experimental group based on promoting the home preparation of meals. These findings provide robust evidence on the effectiveness of the Mediterranean culinary intervention in improving family health and in inducing economic benefits, especially in socioeconomically disadvantaged settings, where food expenditure can be a significant burden.

Schepers and Annemans (2018) [30], examined the impact of Mediterranean and soy-based diets on health and economy, assessing their effects over a 20-year time frame. Using a decision model, they predicted the health outcomes and costs associated with these diets, adopting a societal perspective. Results indicated that both diets result in significant health benefits and economic savings. The soy-based diet was associated with a reduction in cancer prevalence of 0.93–1.57% in women and a reduction of 0.24–1.45% for stroke in men, with projected societal savings of EUR 2,146,000 in Belgium and EUR 1,653,000 in the United Kingdom. Similarly, the DM led to a 1.04–1.34% reduction in diabetes in women and men, with projected social savings of EUR 1,618,000 in Belgium and EUR 1,595,000 in the UK. However, the analysis pointed out some limitations, such as the lack of data specific to the Belgian population regarding certain diseases and the assumption that diet adherence levels remain constant over time. In addition, the data on soy diets came mainly from studies conducted in Asia, which may limit the generalizability to the Western population. In summary, the article provided evidence on the effectiveness of Mediterranean and soy diets in improving health and reducing social costs but suggested the need for further research to confirm these findings and evaluate the effectiveness of any interventions to promote these diets.

Schröder et al. (2016) [31] examined the association between the monetary cost of diet and adherence to the DM in a large sample of young Spaniards. Using data from the enKid project, a cross-sectional survey of a broad spectrum of Spanish individuals aged 2 to 24 years, they found that adherence to the DM, as assessed by the KIDMED index, was positively associated with the monetary cost of the diet, both in terms of daily cost and cost per 1000 kcal per day. In addition, participants with higher adherence to the DM seemed to spend more on foods such as fish, dairy products, and fruits/vegetables than those with low adherence. The average daily monetary cost of the diet was estimated at EUR 3.16 per day, with higher expenditure in males than females. In addition, higher adherence to the DM resulted in an increase in the daily cost of the diet of approximately EUR 0.71. These results underscore the importance of higher diet quality, as measured by adherence to the DM, being associated with higher monetary costs of the diet in young Spaniards. This association is of particular importance considering that young people from disadvantaged socioeconomic backgrounds tend to have a lower quality diet at a lower cost. Therefore, ensuring affordability of healthy foods may be a priority goal to promote healthy eating in the younger generation.

In the same year, the author investigated the relationship between variations in individual diet cost and diet quality and their impact on body weight in a representative population of Girona, Spain [32]. Using data collected between 2000 and 2009, it was observed that an increase in diet cost was significantly associated with improved diet quality and more effective body weight management. Specifically, an increase in diet energy cost was correlated with a decrease in average body weight by 0.3 kg and BMI by 0.1 kg/m^2^ for each EUR 1 increase. These associations were also confirmed through linear regression model analysis, where positive changes in diet cost were correlated with a substantial improvement in adherence to the DM and a reduction in dietary energy density. In economic terms, the results indicate that the high-quality diet results in an average daily cost of about EUR 2.95 (USD 3.33) more than a low-quality diet, equivalent to an annual increase in food expenditure of about EUR 1076 (USD 1215) per individual. This highlights the possible financial challenge that low-income households may face in pursuing healthier diets. In conclusion, the study suggests that targeted food and tax policies, such as subsidies for healthy foods and taxes on less healthy foods, could be crucial in making healthy diets more accessible to all income groups. These interventions could help promote public health and reduce the risks of diet-related diseases.

The study conducted by Pastor et al. (2021) [33] aimed to examine the relationship between adherence to the DM and the monetary cost associated with the diet among a cohort of Spanish schoolchildren aged 6 to 12 years, residing in the province of Avila. Through a cross-sectional survey involving 130 participants, detailed data on dietary habits and associated costs were collected. The results show a significant correlation between DM adherence and the monetary cost of the diet. Specifically, higher DM adherence was found to be associated with a significant increase in diet cost, quantified as an average increase of EUR 1.93 per 1000 kcal consumed. This finding highlights the financial impact of adopting healthier dietary patterns, such as DM, and underscores the need for economic considerations in promoting healthier eating habits, especially in vulnerable socioeconomic settings. These findings provide a scientific basis for the design of interventions and public policies aimed at ensuring affordability of healthy food options for all segments of the population, thereby helping to improve nutritional health and reduce socioeconomic disparities in nutrition.

The study conducted in Belgium by Pedroni et al. (2021) [34] reveals that high-quality diets are associated with an average daily food cost without alcohol of about USD 6.51 or EUR 5.79. This cost increases significantly as diet quality increases: for example, the average daily cost between the lower and upper third of the Healthy Diet Index (HDI) score differed by 14.5 percent, equivalent to about USD 1.04 or EUR 0.92. In addition, the association between diet quality and cost was more pronounced among individuals with lower levels of education, with a significant increase in the cost associated with diet quality in this group. These results indicate that low-income individuals may face a financial challenge in meeting the cost of a healthy diet. Consequently, the implementation of public policies aimed at reducing disparities in diet quality and increasing affordability of healthy foods is recommended. This could include the introduction of tax incentives to make healthy foods more affordable and policies that promote nutrition education and improved access to healthy foods in low-income communities.

Rubini et al. (2022) [35] delves into the link between the DM and CVD, with a focus on its economic impact and adherence in Spain’s Extremadura region of low per capita income. The DM is widely recognized for its cardioprotective benefits, but its cost and affordability remain understudied. Using a representative sample of 2833 individuals, the average monthly food cost associated with the DM was estimated at EUR 203.63, showing a positive correlation between cost and DM adherence. In addition, significant results emerged indicating a higher cost for men, those aged 45–54 years, and those residing in urban areas. The article emphasizes the importance of public policies in improving diet quality, especially in the most disadvantaged socioeconomic groups, suggesting the implementation of educational strategies on nutrition and the adoption of targeted fiscal measures to make DM more affordable. In summary, the paper offers a clear perspective on the importance of DM as an integral part of CVD prevention, while considering the economic implications in adopting healthy eating habits.

The article by Gualtieri et al. (2023) [15] analyzed the eating habits of Italians during the COVID-19 pandemic, focusing on the DM and its costs. According to the reported data, those following the DM were shown to spend less on food than those following less healthy diets. During the lockdown period, the average weekly expenditure on food was found to be significantly different among different levels of DM adherence. For example, those with high DM adherence spent on average about 20 percent less than those with low adherence. In detail, individuals with high adherence spent about 47.70 euros per week, while those with medium or low adherence spent about 58.30 euros per week. Interestingly, as DM adherence increased, the cost of spending decreased. In addition, there was an additional cost involved in purchasing organic foods within the DM, but despite this, consumers who purchased organic foods showed higher adherence to the DM. Specifically, 75 percent of those who purchased organic foods reported high or medium adherence to the DM, compared to 55 percent of those who did not purchase organic products. These data indicate that adopting the DM not only promotes health but can also lead to significant savings on weekly food expenditures. Promoting a DM could therefore be not only an investment in public health, but also a beneficial economic strategy for Italian families.

## 4. Discussion

This study was conducted to update a 10-year-old review with the aim of examining and evaluating new evidence concerning the health–economic implications of adhering to the Mediterranean model. The included studies highlighted the links between dietary habits, health, and economic aspects related to the cost of health care spending.

The papers by Jones et al., Schepers and Annemans, Lampropoulos et al., and Segal et al. [23,24,25,30] provide important assessments of the savings, and consequently the gain, on health care expenditures incurred by high DM adherence. The first two studies estimate direct and indirect savings on health care expenditures using CVD and type 2 diabetes as outcomes. In both cases, predictive models are used to estimate health expenditure savings. Jones et al. [23], assuming an adherence to the Mediterranean model by 20 percent of the population, over and above those already adhering, in one year, estimates a reduction in health care spending in the United States of USD 8.2 billion and in Canada of CAD 0.32 billion. Assuming an 80% increase in adherence, the savings would be USD 31 billion and CAD 1.2 billion. Calculating 10-year cumulative savings, these could range from USD 157.1 billion to USD 596.1 billion in the United States and CAD 5.5 billion to CAD 20.9 billion in Canada. In Schepers and Annemans [30] model, adherence to the Mediterranean model can reduce type 2 diabetes by 1.04–1.34% in women and men, with projected societal savings of EUR 1,618,000 in Belgium and EUR 1,595,000 in the UK. Assuming adherence of 10% of the population for 20 years the estimated savings are EUR 1.55 billion in Belgium and GBP 7.53 billion in the UK.

Lampropoulos et al. [24] and Segal et al. [25] do not use models but both carry out work on patients. The former investigates the economic benefits on length of hospitalization on elderly patients adherent or not to the Mediterranean model. Hospital stays decreased by 0.3 days for each unit increase in DM score (*p* < 0.0001), by 2.1 days for each 1 g/dL increase in albumin (*p* = 0.001), which directly correlated with good nutritional status. Mortality risk decreased by 13% for each unit increase in DM score (HR, 0.87; *p* < 0.0001). Segal et al. evaluated a DM implementation intervention in patients with major depression. The author recruited adults and randomly assigned them to either a group dietary program or a social program. The dietary group had counseling followed by Mediterranean-style group cooking meetings; the social group attended meetings. Both groups showed improvements in symptomatology but the group that implemented a Mediterranean lifestyle achieved greater improvements, which were maintained even at 6-month follow-up. The dietary intervention is highly cost-effective, with a cost per QALY of about EUR 1775, well below the threshold considered acceptable in health care.

Considering DM adherence and cost of expenditure, we positively valued the inclusion of studies with mixed results. The work of Seconda et al., Tong et al., Pastor et al., Pedroni et al., Rubini et al., and both papers by Schröder et al. [23,26,27,31,32,34,35] agree in their findings associating increased DM adherence with increased food procurement costs.

Contextually, the works of Schröder et al., Pedroni et al., and Rubini et al. [31,34,35] show a positive association between low socioeconomic status and lower adherence to a DM, due to both economic and cultural factors. In Schröder et al.’s [31] study of a sample of students in Girona, a Spanish university town, it becomes evident how the reduced economic availability of the off-site students affects their food choices.

Midway stands the work of Yacoub Bach et al. [28], who, by analyzing different dietary types, show that as animal sources within the diet are reduced, the savings are greater in economic terms. Moreover, it is fundamental to recall that the DM is characterized by a high consumption of vegetables, fruits, whole grains, legumes, and oil and only moderate intakes of foods of animal origin. It therefore turns out to be cost-effective compared to typical Western diets. Furthermore, from a nutritional point of view, diets with a higher content of foods of non-animal origin are more complete.

Instead, in the studies of Lara et al., Razavi et al., and Gualtieri et al. [15,22,29] neutral or inverse associations of a DM and cost of groceries were found. Lara et al., by implementing an educational intervention on two different groups, found a significant improvement in DM adherence in both, together with no significant increase in the expenditure cost. Most likely, as reported by the author, the participants had reduced meat consumption by preferring plant sources.

Razavi et al. [29], by implementing a DM cooking and education course in family groups, found increased adherence to the Mediterranean lifestyle compared to the control group. The increased adherence was significantly associated with savings on weekly food costs in households from increased consumption of home-cooked meals, avoiding ready-to-eat food from the restaurant industry. The most interesting research that also recognizes the DM as saving on groceries is the study by Gualtieri et al. [15]. In fact, it is significantly evident that high adherence to a DM reduces the cost of groceries. The latter, moreover, decreased as adherence to the Mediterranean model increased. This very study, the only Italian one, estimates the total cost spend. The previous studies cited estimate the total cost spent considered cost spending by calibrating it to 1000 kcal. It seems clear that this assessment has inherent limitations, since Mediterranean food is much less calorically dense than food far from the Mediterranean model.

Finally, some studies use predictive models to estimate health expenditure savings from DM adoption. These models consider both direct savings, such as reduced costs associated with medical care for CVD and type 2 diabetes, and indirect savings, such as improved quality of life and greater economic productivity due to better health. However, the magnitude of these savings depends on actual adherence to the DM and the duration of observation.

Other studies conducted on elderly patients with major depression provide strong evidence on the beneficial effects of the DM on health. A significant reduction in hospital stay and mortality risk is observed in elderly patients following a DM. In addition, the adoption of a DM appears to contribute to improvements in depressive symptomatology, with potential savings in health care spending associated with the management of depression-related health conditions.

However, some studies also point to some economic disadvantages associated with DM adoption. For example, there is an increase in food procurement costs, a factor that, in low-income populations, could affect the difficulty in accessing ingredients typical of the DM. These findings suggest the need to also consider socioeconomic factors when promoting a DM and assessing its overall economic impact.

Other studies, however, indicate that the adoption of a DM could result in overall savings on weekly food expenditures, both through increased home meal preparation but especially through the consumption of less expensive foods. In fact, in a DM, we find most food sources derived from the plant world and only sporadic consumption of animal sources. Moreover, the Mediterranean model requires the use of products that follow seasonality, which is a determining factor in lowering cost and environmental sustainability.

Our update of the previous review has similar limitations to the first version. The main limitations are related to the scarcity of studies assessing the relationship between food costs and adherence to different dietary patterns. In addition, the selected studies used different types of food frequency questionnaires, so the score for adherence to the DM was calculated in different ways. Other elements that allow for only a partial comparison of the studies are related to the different geographical contexts, the dissimilar characteristics of the participants, etc. The small number of economic studies remains worrying considering the importance of diet as a risk factor for the incidence and progression of common chronic diseases and obesity. Another limitation concerns the quality of the studies and data supporting the modelling of the relationships between intermediate outcomes and health. It remains desirable for interventions to be compared with all other likely and available treatment options in order to assess the true cost-effectiveness of the intervention.

## 5. Conclusions

The conclusions drawn from the texts indicate a clear link between dietary habits, health, and economic aspects related to dietary cost and health spending. The studies reviewed provide an overview of the potential impact of adopting the DM on these factors, showing mixed but overall promising results.

In conclusion, recognizing the significant health benefits associated with adopting a DM and the potential savings on health care spending, it is important to consider policies that support this lifestyle. National governments should seriously take into account the emerging data and focus on population educational programs and prevention campaigns. Such policies would return a gain beyond the mere economic aspect, which is still more than considerable. In addition, it is essential that governments implement beneficial economic measures for foods in line with the DM and place measures to discourage the purchase of non-Mediterranean foods. Special attention should be paid to the regulation of advertising and marketing campaigns, especially to protect children. Further research is needed to fully assess the long-term health and economic impact of the DM and to develop effective strategies to promote it in an equitable and sustainable manner. 

## Figures and Tables

**Figure 1 nutrients-16-01899-f001:**
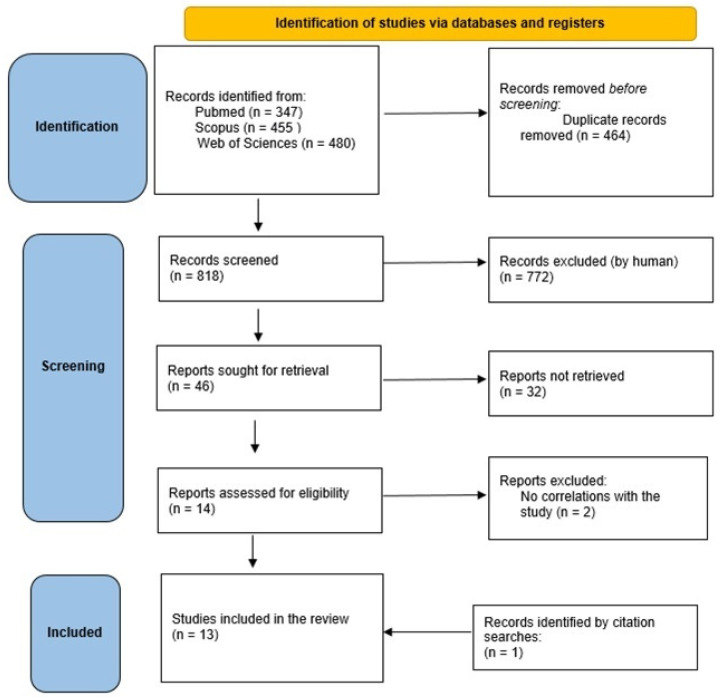
PRISMA flowchart of the selection process.

**Table 1 nutrients-16-01899-t001:** Characteristics of the study included in the systematic review.

References	Study Design	Type of Analyses	Diseases Outcomes	Alternatives	Nation/Perspective	Sample	Efficacy Measures/ Cost Measures	Main Results	QA (NOS) or (RCT)
Lara et al., 2015 [22]	RCT	Cost of the diet	No	Evaluation of the comprehension, acceptability, feasibility, adherence, and cost of the DM in an elderly population	England/elderly population	23 healthy men and women aged 50 and over	Direct Costs	The average daily cost of food intake, estimated from the 3-day food diaries, was not significantly different between pre- and post-intervention for group 1, group 2, or the combined sample. The daily food cost was not significantly different between groups 1 and 2. Linear regression analysis showed that a one-point increase in the DM score represented a cost of 0.55 pounds. Furthermore, participants were asked to report their perception of any difference in the cost of their diet before and after the intervention. Compared to the perceived cost of their usual diet before the intervention, the vast majority (87%) of participants reported that the perceived cost of their diet after 3 weeks of intervention was less than or equal to that of their usual diet.	4
Jones et al., 2019 [23]	Cost-of-illness analysis	Economic evaluation	CVD	Direct and indirect economic advantages of high adherence to DM over CVD	Canada/United States	No	Cost of illness	Increasing the proportion of the population adhering to DM by 20% above the current level of adherence resulted in annual savings in CVD-related costs of USD 8.2 billion (95% confidence interval [CI]: USD 7.5–8.8 billion) in the United States and CAD 0.32 billion (95% CI: CAD 0.29–0.34 billion) in Canada. An 80% increase in adherence resulted in savings of USD 31 billion (95% CI: USD 28.6–33.3 billion) and CAD 1.2 billion (95% CI: CAD 1.11–1.30 billion) in each country.	
Lampropoulos et al., 2020 [24]	Cost-of-illness analysis	Evaluation of dietary adherence effectiveness	Length of hospital stay	Economic advantages over hospitalization of patients with high DM adherence	Greece	183	Hospitalization costs	The length of hospital stay decreased by 0.3 days for each unit increase in the DM score (*p* < 0.0001), by 2.1 days for each 1 g/dL increase in albumin (*p* = 0.001) and by 0.1 days for each day of previous hospitalization (*p* < 0.0001). Prolonged hospitalization (*p* < 0.0001) and its interaction with DM score (*p* = 0.01) remained the significantly associated variables for financial cost. Mortality risk increased by 3% for each year of age increase (hazard ratio [HR], 1.03; *p* = 0.02) and 6% for each previous admission (HR, 1.06; *p* = 0.04), and decreased by 13% for each unit of DM score increase (HR, 0.87; *p* < 0.0001).	9
Segal et al., 2020 [25]	Economic evaluation	Utility and cost-effectiveness analysis of the dietary intervention	Depression	Utility and cost-effectiveness analysis of a Mediterranean dietary intervention in the treatment of depression	Australia	152	- Direct costs - QALY (quality-adjusted life year)	An intervention that implemented the DM (including cooking workshops) in persons diagnosed with major depression was found to be extremely cost-effective in curing the condition compared to a group in which the intervention was based on social activities.	
Seconda et al., 2017 [26]	Cohort	Utility and cost-effectiveness analysis of different dietary models	No	Comparison of four dietary models assessing Mediterranean adherence, sustainability, health, and cost	France	22,866	Direct/indirect costs	This study showed that sustainability indicators related to health and nutrition, environmental impact, and sustainability for purchasing and sociocultural aspects were consistently better among Conv-Med, Org-NoMed, or Org-Med participants than among Conv-NoMed participants. Furthermore, it is interesting to note that the combination of both the DM and the high consumption of organic food (Org-Med group) was associated with better values of the indicators related to sustainability, apart from the cost of the diet.	6
Tong et al., 2018 [27]	Cohort	Cost of the diet	No	Association between diet cost and adherence to the DM in a non-Mediterranean country	England	12,435	Direct costs	High adherence to the DM was associated with higher dietary costs. On average, high adherence to the DM (mean dietary cost: GBP 4.47, 95% CI 4.44, 4.49) was associated with a price difference of GBP 0.20 per day (95% CI 0.16, 0.24) compared to low adherence (GBP 4.26, 95% CI 4.23, 4.29), equivalent to 5.4% (95% CI 4.4, 6.4)	3
Yacoub Bach et al., 2023 [28]	Observational study	Cost and dietary sustainability	No	Analysis of sustainability, nutritional quality, and cost among different diets	N/A	No	- CO_2_ emissions–land use - Water withdrawals - Nutritional quality - Economic accessibility	The study shows that vegan, Mediterranean, and vegetarian diets are the most sustainable in all parameters, while diets rich in meat have the greatest negative environmental impact. Diets based on WHO dietary guidelines performed poorly in terms of convenience, environmental impact, and nutritional quality. Diets with higher nutritional quality included the vegan, paleo, and DM. Diets that eliminated meat were the cheapest in terms of both total cost and cost per gram of food.	/
Razavi et al., 2020 [29]	RCT	Cost and environmental sustainability of the diet	No	Effects of a culinary education program on diet cost and adherence to the DM	United States	41 families	Direct costs	Households participating in hands-on kitchen-based nutrition education were almost three times more likely to follow a Mediterranean dietary pattern (OR 2–93, 95% CI 1–73, 4–95; *p* < 0.001), with a 0.43 point increase in adherence to the DM after 6 weeks (B = 0.43; *p* < 0.001), compared to those with traditional nutrition counseling. Furthermore, kitchen-based nutrition education projects saved families USD 21.70 per week compared to standard dietary counseling, increasing the likelihood of eating home-prepared meals compared to commercially prepared meals (OR 1.56, 95% CI 1.08, 2–25; *p* = 0–018).	2
Schepers and Annemans 2018 [30]	Prediction model study	Cost-effectiveness analysis	Chronic Diseases	Evaluation of health and economic effects of plant-based diets	Belgium/England	1000 men and 1000 women	Direct/indirect costs	Based on the study’s model, if 10% of the total population were to strongly commit to the DM, the social cost savings in Belgium and the UK over 20 years would be estimated at EUR 1.55 billion and GBP 7.53 billion, respectively.	/
Schröder et al., 2016 [31]	Cross-sectional	Cost of the diet	No	Relationships between diet cost, socioeconomic status, and adherence to the Mediterranean model (students)	Spain	1629 boys and 1905 girls	Direct costs	Socioeconomic status was positively associated with daily diet cost and diet quality as measured by the KIDMED index (EUR/day and EUR/1000 kcal/day, *p* < 0.019). High adherence to the DM (KIDMED score 8–12) was EUR 0.71/day (EUR 0.28/1000 kcal/day) more expensive than low adherence (KIDMED score 0–3). The higher daily cost of diet is associated with healthy eating in young Spaniards. Higher socioeconomic status is a determinant factor for higher daily diet cost and quality.	8
Schröder et al., 2021 [32]	Prospective, population-based study	Cost of the diet	Obesity and overweight	Relationship between diet cost, food quality, and weight loss (adults)	Spain	2181	Direct costs	The average daily diet cost increased from 3.68 (SD 0.89) EUR/8.36 MJ to 4.97 (SD 1.16) EUR/8.36 MJ during the study period. This increase was significantly associated with improved diet quality (Δ ED and Δ DMS-rec; *p* < 0–0001). Each EUR 1 increase in the monetary cost of the diet per 8.36 MJ was associated with a 0.3 kg decrease in body weight (*p* = 0.02) and 0.1 kg/m^2^ in BMI (*p* = 0.02). An improvement in diet quality and better weight management were both associated with an increase in diet cost	9
Pastor et al., 2021 [33]	Cross-sectional	Cost of the diet	No	Relationship between cost and diet quality (children)	Spain	130 children	Direct costs	A direct relationship was observed between diet cost and DM adherence [OR (EUR/1000 kcal/day) = 3.012; CI (95%): 1.291; 7.026; *p* = 0.011]. It should be noted that the cohort examined had low overall adherence to the DM.	8
Pedroni et al., 2021 [34]	Cross-sectional	Cost of the diet	No	Cost differences between different dietary quality and sociodemographic characteristics	Belgium	1158	Direct costs	The mean cost of the daily diet was USD 6.51 (standard error of the mean [SEM] USD 0.08; EUR 5.79 [EUR 0.07]). Adjusted for covariates and energy intake, the mean daily diet cost (SEM) was significantly higher in the highest tercile (T3) of both diet quality scores compared to T1 (DM score: T1 = USD 6.29 [USD 0.10]; EUR 5.60 [EUR 0.09] vs. T3 = USD 6.78 [USD 0.11]; EUR 6.03 [EUR 0.10]; Healthy Diet Indicator: T1 = USD 6.09 [USD 0.10]; EUR 5.42 [EUR 0.09] vs. T3 = USD 7.13 [USD 0.11]; EUR 6.34 [EUR 0.10]). Both diet quality and cost were higher in respondents aged 35–64 (compared to those aged 18–34), working (compared to students), and those with higher levels of education. The association between quality and cost of diets was weaker in men and among those with higher levels of education.	9
Rubini et al., 2022 [35]	Cohort	Cost of the diet	No	Cost and adherence to the Mediterranean model in a low-income region	Spain	2833	Direct costs	The average monthly cost was EUR 203.6 (IQR: 154.04–265.37). Food expenditure was higher for men (*p* < 0.001), for the 45–54 age group (*p* < 0.013), and for those living in urban areas (*p* < 0.001). A positive correlation was found between food-related expenditure and adherence to the DM. The median monthly cost represented 15% of the mean disposable income, varying between 11% for the group with low adherence to the DM and 17% for the group with high adherence to the DM. The monthly cost of the DM was positively correlated with the degree of adherence to this dietary pattern.	3
Gualtieri et al., 2023 [15]	Cross-sectional	Cost and environmental sustainability of the diet	No	Adherence to the DM and its effect on health and environmental and socioeconomic sustainability during the COVID-19 pandemic in an Italian population sample	Italy	3353	- CO_2_ emissions - Water consumption - Food cost - Adherence to the Mediterranean model	The low and medium adherence groups showed higher CO_2_ and a higher H_2_O emissions than the high adherence group (*p* < 0.001 and *p* < 0.001, respectively). Similarly, the medium adherence group had higher CO and higher H_2_O than the low adherence group (*p* < 0.001). In addition, a lower BMI was associated with a decrease in CO_2_ and H_2_O emissions. Food costs resulted in statistically significant differences between the MEDAS groups. In fact, the high adherence group had a lower weekly food cost than the low and medium adherence groups (*p* < 0.001 and *p* < 0.001, respectively), and the medium adherence group had a lower weekly food cost than the low adherence group (*p* = 0.004).	10

**Table 2 nutrients-16-01899-t002:** Quality of the cost-effectiveness analysis of the included studies (according to Drummond’s checklist, modified by La Torre et al. [21]).

Authors (Year of Publication)	Schepers and Annemans 2018 [30]	Segal et al., 2020 [25]	Jones et al., 2019 [23]
Study design Score/items (25 score items)	23/25	24/25	16/25
Data collection Score/items (46 score items)	36/46	35/46	45/46
Analysis and interpretations of results Score/items (45 score items)	36/45	45/46	36/45
Total score/items	95/116 = 81.89%	104/116 = 89.65%	97/116 = 83.62%

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
