# Peer review of "Cost and Cost-Effectiveness of the Mediterranean Diet: An Update of a Systematic Review"

_nutrients, 2024, doi:10.3390/nu16121899_

Round 1

Reviewer 1 Report

Comments and Suggestions for Authors

TO THE AUTHORS

Brief Summary

This study was a rigorous comprehensive review of the existing bibliography exploring the sustainability, cost-effectiveness, and health economic impact of compliance (low and high) to the Mediterranean Diet model. Assessment of 15 relevant studies revealed that high adherence to the MD was associated with a reduction in both health and economic costs. Overall, data analysis revealed that dietary costs were proportional to diet quality. Adherence to the MD pattern was associated with a slight increase in food costs but substantial improvements in individual and family health along with household and national economic savings. In comparison to low adherence to MD, higher adherence was associated with higher expenditure for fish, dairy products and fruits/vegetables, which could incur economic burden in low SES groups. In conclusion, this review provides evidence on the effectiveness of MD in improving health and reducing social economic costs. Considering the cost of adherence to, the MD is pivotal in adoption of this high-quality dietary pattern and has significant implications for public health and the healthcare system. Affordability of healthy foods may be a priority goal to promote healthy eating and ensure adherence to the MD model in low SES groups.

The Mediterranean diet (MD) is widely recognized for its protective benefits from future chronic disease, but its sustainability in terms of cost and affordability remain understudied.  This systematic review of the literature provided insight into the link between dietary habits, health, and the economic aspects related to the cost, healthcare expenditure and sustainability of the Mediterranean diet.

Overall, this manuscript was well-structured and is a scientifically-sound study, enabling transparency and replication of results. The authors demonstrate that they have mastered the art of writing systematic reviews for scientific journals. Well-done!

However, there are a couple of minor points that require revision which will improve the manuscript.

Please refer to my comment below. We look forward to more studies on the MD from this research group.

Comments

Abstract

Line 21 Define DM in the abstract

Line 22 life-style. Remove hyphen.

Conclusion

Line 508-530 ‘Some studies use predictive models to estimate health expenditure savings from DM adoption. These models consider both direct savings…………………….. Moreover, the Mediterranean model requires the use of products that follow seasonality, which is a determining factor in lowering cost and environmental sustainability.

-Study limitations should be moved to the discussion and the conclusions summarized in brief, corresponding to the conclusion of the abstract.

Reviewer 2 Report

Comments and Suggestions for Authors

Thank you for conducting this follow-up study to your 2013 publication. Unfortunately, this manuscript will need extensive editing prior to publication. I will begin with some general statements that apply to the entire manuscript and then give specific edits based on the line numbers in the manuscript I was sent.

General comments:

While the various studies that you have cited have used different abbreviations for 'Mediterranean Diet', which you have then used in your descriptions of the publications, the result is that it is challenging for the reader to follow. I highly recommend that you introduce the abbreviation of DM for Mediterranean Diet in line 10 of your abstract and then again in line 33 of the Introduction and then continue to use this abbreviation throughout your manuscript - regardless of what the various authors have used in their manuscripts.

Also, throughout the manuscript whenever you introduce an abbreviation, please also spell it out for your reader. Even abbreviations as commonly used as WHO - need to be spelled out the first time they are used.

Abstract: Please remove the section designations in the abstract, such as "Background", "Methods", etc. they are not typically used in the abstract for this and other journals. This change also gives you more words for your content!

In the section below, the comment is a recommendation on how to reword the text for better clarity for the reader.

Line number     Comment

10     It is well known that the Mediterranean Diet (DM) is beneficial for health, as years of research globally have confirmed. 

14   The research approach utilized 

15    ... bases: PubMed, Scopus, and Web of Science.

15     Please move the sentence "Data extraction and analysis were performed on all included studies." to after the CRD citation in line 17. 

22    add after the word important: " for national public health programs" that...

28    please add the word "globally" into this sentence.

33    please add DM after Mediterranean Diet here and then replace Mediterranean Diet with DM throughout the remainder of the manuscript

34    please add: beneficial "long-term" health effects.

40    add appropriate reference call out for the Ancel Keys paper and add it to the reference list

40    reword: "... has been evolving, however, many of the same key components..."

43    "...processed meats and sweets have been supported."

60     spell out OECD

63    spell out WHO

81    [17], published 10 years ago, with 

100    add sentence that defines and explains what ZOTERO is.

109-116    replace 1 through iv with "a) and b)" in both locations as included and excluded are better not contained in the same list.   

119    Begin this paragraph with the sentence that starts:" A data collection form..." then follow with the rest of the sentences starting with the sentence: "Data extraction...". 

129    Replace "was carried out" with "was conducted".

153    What is meant by the use of the term "strings" here? Please replace these two sentences with a more detailed explanation. 

167    Figure 1 is excellent!

193    Throughout the manuscript, please remove the first names of the authors form the in text citations: Lara Jose et al should be Jose et al.

204    reword to read: "The second study was 3 weeks in duration with two intervention arms."

206    "without incurring significantly higher costs". Remove "albeit not statistically significant".

212    remove the word " in"  to become "... incurring significantly higher costs".

243    change "evaluates" to "evaluated". 

262    change "analyzes" to "analyzed"

for the following line numbers change the verbs all to past tense:

265 "compared"; 267 "followed"; 268 "showed"; 270 "was"; 272 "was"; food "was" 

279  "focused"; 288: "showed"; 

295    deleted the word "considered"

299 change to "tended"

301 changed to "showed"

303    Please define SAIN:LIM, tell what is does,  and provide a reference for it

306    foods "tended"

307    change to read: 'products appeared to be more expensive..."

311    "examined"; 315 "revealed"

319    "... experimental group based on promoting home..."

324     Wherever you cite the Schepers reference please list it as Schepers & Annemans.; change to "examined"

Please change the verbs in the remainder of the paragraphs to past tense as I have indicated above. I recommend that you have someone reread the draft prior to resubmission. 

329    reword and add "a reduction of 0.24-1.45% for stroke in men."

375     the paragraph beginning with this line reads very well!! For the remining paragraphs in the Results section, please be sure to change Mediterranean diet to DM throughout 

433-435    This paragraph is very unclear Please revise and add at least 1 more sentence to explain what is meant.

437  delete "us with"

440   delete "the former" and line 442 delete "the latter

entire paragraph< does bn mean billion? please replace throughout with the full word

450  " for 20"??? for 20 what is meant - years? if so please add

479    please reword to read: " "... as animal sources within the diet are reduced, the savings are greater in economic terms."

499   "estimates the total cost spent".  Delete "above, on the other hand"
